# CardBench: A Benchmark for Learned Cardinality Estimation in Relational Databases

## Abstract

Cardinality estimation is crucial for enabling high query performance in relational databases. Recently learned cardinality estimation models have been proposed to improve accuracy but there is no systematic benchmark or datasets which allows researchers to evaluate the progress made by new learned approaches and even systematically develop new learned approaches. In this paper, we are releasing a benchmark, containing hundreds of thousands of queries over 20 distinct real-world databases for learned cardinality estimation. In contrast to other initial benchmarks, our benchmark is much more diverse and can be used for training and testing learned models systematically. Using this benchmark, we explored whether learned cardinality estimation can be transferred to an unseen dataset in a zero-shot manner. We trained GNN-based and transformer-based models to study the problem in three setups: 1-) instance-based, 2-) zero-shot, and 3-) fine-tuned.

Our results show that while we get promising results for zero-shot cardinality estimation on simple single table queries; as soon as we add joins, the accuracy drops. However, we show that with fine-tuning, we can still utilize pre-trained models for cardinality estimation, significantly reducing training overheads compared to instance specific models. We are open sourcing our scripts to collect statistics, generate queries and training datasets to foster more extensive research, also from the ML community on the important problem of cardinality estimation and in particular improve on recent directions such as pre-trained cardinality estimation.

## 1 Introduction

*Cardinality estimation* (CE), the problem of determining the number of intermediate records that will be returned by a database query, is one of the key building block in modern databases for optimizing performance, and has been studied extensively by the database research community. CE is used by query optimizers to choose between alternative query execution plans. A query plan that produces the smallest intermediate results and processes the least amount of data usually leads to better execution times, and hence is preferred. Thus CE is critical for optimization decisions such as choosing performant join orders, deciding on whether or not to use an index, and picking the best join method. It has been shown (10) that "bad" cardinality estimates lead to "bad" execution plans that can have multiple orders of magnitude worse performance. CE is also critical in database recommenders to choose between alternative indexes, materializes views, and storage layouts.

Traditional CE techniques used in modern database systems have well-known limitations, as they make simplistic data modeling assumptions, such as data *uniformity* and *independence* of columns in the tables. Learned CE model following an instance-based approach have been shown to improve upon the state-of-art techniques used by database systems today (9; 7; 22; 8; 19; 15). Despite their better accuracy, learned CE models have not been adapted in practice, due to their high training overheads, among other reasons. As a result, pre-trained CE models, which are trained on a corpus of diverse datasets on transferable features, are highly desirable to alleviate the training costs and increase adaption.

**Our Main Contributions.**    In this paper we release **CardBench**, a benchmark containing thousands of queries on 20 distinct databases, and scripts to compute data summary statistics and generate queries. We generated three training datasets; one with queries over single tables with multiple filter predicates, one with queries with a binary join between two tables and multiple filter predicates on

each table and one with queries with multiple binary joins and multiple filters predicates on each table. Further, we release a query generator and flexible infrastructure that can be used and extended to generate even more complex training queries. In comparison to existing CE benchmarks, CardBench includes a much higher number of datasets and training data (i.e., queries and cardinalities) to experiment with and compare different types of learned CE approaches.

To show the benchmark in-action, in this paper we have trained GNN-based and transformer-based CE models, under 3 setups: 1-) instance-based (i.e., a CE model trained only for the particular dataset), 2-) pre-trained CE model in zero-shot mode, 3-) pre-trained CE model with fine-tuning on the dataset. Our results show that although CE is harder to learn with a pre-trained model in a zero-shot setting, but with a small amount of samples fine-tuning a pre-trained model can achieve high accuracy as good as, if not better than the instance-based methods. We hope that releasing CardBench to the public community will help to reduce the barrier-of-entry for experimentation with learned CE models and also foster new research, also from the ML community, and lead to further advances on this highly important problem.

## 2  BACKGROUND AND RELATED WORK

**Learned Cardinality Estimation.** Traditional CE methods rely on heuristics and simple analytical models (5) (e.g., one-dimensional histograms or sketches). Recently, there have been various approaches for learned cardinality estimation (9; 7; 22; 8; 19; 15; 14; 26; 21). While early learned approaches used classical regression models, more recently deep learning based approaches (9; 13) have been proposed which show highly promising accuracy over traditional approaches. All of these methods are instance-based: they utilize instances-specific features like names of tables and columns; they are evaluated and trained on the same database. There are two main approaches: workload driven and data driven. Workload driven approaches need to run a representative set of queries (10 to 100 thousands) to collect true cardinalities as labels for the training samples, incurring a very high overhead. These models also need to be re-trained as the representative workloads shift. To avoid this high overhead of running thousands of queries to collect the true cardinalities per dataset, more recently so called data-driven learned cardinality estimators have been proposed (7; 23). In contrast to the workload-driven approaches, data-driven approaches learn the multi-variate data distribution within and across tables without running any query plan but simply by learning data distributions of the base tables. Data driven approaches still require one model per dataset and need to be re-trained as the data gets updates, which still incurs training overheads. As such, very recently some first generalizable cardinality estimators (12) have been proposed that can be used on datasets in a zero-shot manner; i.e., the model can be used without the need to learn one instance-specific model per dataset. Moreover, zero-shot models also promise that they can automatically provide accurate estimates even if the underlying data of the dataset is updated (i.e., new rows are added or existing ones are updated or deleted). The core idea is that such zero-shot models are pre-trained on a broad spectrum of different datasets and use transferable features, such as table sizes, allowing them to generalize to unseen datasets. We argue that more research is needed on pre-trained models on more compelx queries. For this purpose, we provide a very first large-scale training corpus to foster research in this direction.

**Cardinality Estimation Benchmarks.** To benchmark cardinality estimation models, the early papers have mainly relied on either home-grown benchmarks with non-public datasets or they have used some available benchmarks that have been originally designed for other purposes. One central benchmark that has been used so far extensively by many papers is the Join Order Benchmark (JOB) (11), which is based on real-world data from IMDb, containing interesting data distributions and correlations. JOB contains 79 queries, which challenge cardinality estimation. This is in stark contrast to many other database benchmarks such as the TPC benchmarks which use synthetic data that is way simpler and thus does not represent a real challenge for cardinality estimation. However, one benchmark with one dataset is not sufficient to benchmark learned cardinality estimation and show its robustness towards various different real-world domains. As such, in the recent years, there have been further efforts to create additional benchmarks that can be used to test cardinality estimation models. For example, (4) proposes a new benchmark, which contains one new real-world dataset STATS and a generated query workload that is more diverse in terms of query and data characteristics than JOB. Another benchmark (13), which aims in a similar direction uses two different datasets - IMDB and

Table 1: Database Statistics. All statistics consider all tables of a database.

| Database Name | # Tables | # Columns | # Rows | # Unique Vals | # Join Paths | Join Type | Correlation |
|---|---|---|---|---|---|---|---|
| accidents | 3 | 43 | 27.4M | 1.8M | 2 | chain | 0.37 |
| airline | 19 | 119 | 944.2M | 123.4K | 27 | star | 0.07 |
| consumer | 3 | 24 | 105.3M | 13.9M | 2 | chain | 0.37 |
| employee | 6 | 24 | 48.8M | 3.8M | 5 | star | 0.02 |
| movielens | 13 | 48 | 78.9M | 5.5M | 6 | mixed | 0.25 |
| (∼)cms_synthetic_patient_data_omop | 24 | 251 | 32.6B | 2.5B | 22 | star | 0.16 |
| (∼)covid19_weathersource_com | 4 | 52 | 34.6B | 11.6M | 2 | chain | 0.41 |
| (∼)crypto_bitcoin_cash | 2 | 27 | 2.0B | 280.4M | 2 | chain | 0.14 |
| (∼)ethereum_blockchain | 7 | 84 | 5.6B | 641.7M | 10 | mixed | 0.23 |
| (∼)geo_openstreetmap | 16 | 81 | 8.3B | 3.2B | 15 | star | 0.11 |
| (∼)github_repos | 9 | 41 | 1.7B | 280.8M | 8 | mixed | 0.00 |
| (∼)human_variant_annotation | 26 | 501 | 7.2B | 1.3B | 17 | star | 0.29 |
| (∼)idc_v10 | 19 | 1660 | 40.0B | 455.7M | 17 | star | 0.07 |
| (∼)open_targets_genetics | 13 | 268 | 17.3B | 392.9M | 21 | mixed | 0.06 |
| (∼)samples | 8 | 273 | 7.0B | 470.9M | 3 | mixed | 0.07 |
| (∼)stackoverflow | 14 | 187 | 3.0B | 1.1B | 13 | star | 0.23 |
| (∼)umiami_lincs | 7 | 52 | 21.8B | 2.2M | 4 | star | -0.04 |
| (∼)usfs_fia | 11 | 1042 | 16.9B | 333.5M | 18 | mixed | 0.05 |
| (∼)uspto_oce_claims | 12 | 88 | 3.5B | 428.1M | 16 | mixed | N/A |
| (∼)wikipedia | 25 | 101 | 33.7B | 514.5M | 43 | mixed | N/A |
| tpch_10G | 8 | 61 | 1.2B | 118.8M | 7 | mixed | 0.02 |

StackExchange (SE) - and contains 16K unique queries and true cardinalities. However, these two benchmarks still mainly target instance-specific models which can be seen by the fact that they only contain one or two different datasets, which is clearly no sufficient for testing pre-trained models that require a representative set of different datasets. In this paper, we thus present a benchmark that targets the pre-trained cardinality estimation and thus includes a much more diverse set of 20 different datasets which can be used for pre-training and testing.

## 3 CARDBENCH

for CE, that contains 20 distinct datasets and thousands of queries of different complexities. Our goal is to foster further research in the area of learned CE, with a focus on enabling the training and testing of pre-trained zero-shot CE models but also instance-specific CE as well as fine-tuning pre-trained CE models.

**Dataset Selection:** Table 1 list the CardBench datasets. The datasets were chosen to be diverse, complex and cover a wide range of data distributions to stress CE models. All the datasets are publicly available or are based on publicly available datasets (we list the sources in the CardBench repository). Datasets marked with (∼) in Table 1 are created by random sampling the original datasets to reduce their size. Smaller datasets reduce the cost of statistics calculation and running queries. For example, the size of the original *github_repos* dataset is 3 petabytes, using random sampling the version we use in CardBench is 300 gigabytes in size. The number of datasets included with CardBench was chosen to allow for training zero-shot models, which is validated by our experiments and previous work (6). In Table 1 correlation is the column pairwise Pearson correlation for columns of the same table, averaged for all tables. N/A correlation means that all columns of the tables are non numeric, for which a Pearson correlation is not defined.

**Queries and Training Data:** Our training data are SQL queries represented as graphs annotated with dataset statistics and the query cardinality as the label (Table 2, right). CardBench includes three sets of training data, *Single Table*, *Binary Join* and Multi-Join. *Single Table* queries filter a single table using 1 to 4 filter predicates. *Binary Join* queries join two tables that are also filtered with 1 to 3 filter predicates per table. *Binary Join* queries perform 0 to 8 joins and also apply 0 to 2 filter predicates per table. The three training dataset configurations can demonstrate the challenges of more complex queries on CE models (see Section 5). To obtain the training labels (i.e. the query cardinalities) we used System X[1], executing the queries of the three training data sets required more than 14 cpu time years. Table 4 in the Appendix shows the average, max and standard deviation of the query cardinalities of the queries per database and set of training data, these statistics show the diversity of databases and queries. **As such, releasing the queries with their true cardinalities significantly lowers the entry barrier for research on learned CE models.**

---

[1]System name hidden to preserve anonymity

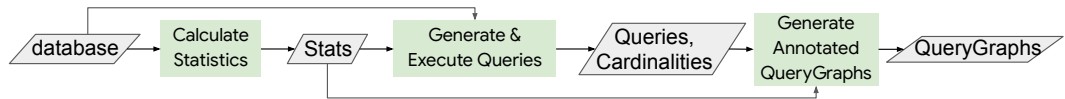

Figure 1: CardBench Benchmark Creation Steps. The scripts to create the benchmark along with the training data (query graphs) released as open-source.

The number of queries we include per dataset varies, the query generator (described in Section 3.1.2) uses dataset information and statistics and follows a random process to create realistic queries. Although we generate the same number of queries per dataset, during query execution we filter out queries that are duplicate, return zero results or timeout while running. As a result, the successfully run queries that become the training data vary per dataset (at minimum we have 5,000 successfully run queries per dataset). We opted not to cap the number of queries per dataset and leave it to the benchmark user to use as many queries as they deem useful. For example, for the experiments shown in Section 5 we use the same number of queries across all datasets to ensure a fair cross-comparison.

**Accuracy Metric** The *q-error* is a common metric in database systems research for evaluating the accuracy of CE models (4; 5; 12). It calculates the relative deviation of the predicted cardinality from the true cardinality for a given query. The Q-error for a single query is calculated as follows:

$$\text{Q-error} = \max\left(\frac{\hat{y}}{y}, \frac{y}{\hat{y}}\right) \in [1, +\infty), \tag{1}$$

where $y$ and $\hat{y}$ represent true and predicted cardinalities, respectively. We particularly focus on P50 and P95 q-errors aggregated across all queries within a dataset to evaluate the models' accuracy in common and tail cases. We refer to Section 5.1.1 for the details of how CardBench is used to train and evaluate zero-shot, instance and fine-tuned CE models.

## 3.1 BENCHMARK CREATION

The process we follow to create CardBench is illustrated in Figure 1 and comprises of three major steps a) statistics calculation, b) query generation and execution, c) annotated query graph generation. The first and second steps of the process involve running SQL queries to calculate statistics and collect the query plans and training labels. Thus, creating CardBench is expensive and for that reason alongside the pipeline that creates the benchmark we release the statistics of the datasets, the queries with their cardinalities and the query graphs.

### 3.1.1 DATASET STATISTICS CALCULATION

In CardBench we include a set of data statistics and summaries which can be used as input features for CE models, we expect traditional query optimizers to calculate and store most of the statistics we use, but as not all query optimizers maintain the same set of statistics or calculate them in the same way, the first step in our infrastructure is statistics calculation. Therefore the statistics used by CardBench are database system agnostic. The calculated statistics are stored in a database and used to create the SQL queries as well as the training data. Table 2 lists the statistics we calculate/collect.

**Primary foreign key relationships** (pk-fk) are part of a database schema design (17). A pk-fk relationship is a logical connection between the columns of two tables semantically linking their information. Naturally we want to use pk-fk relationships when creating synthetic workloads. A challenge we faced is that open source datasets do not usually specify pk-fk relationships, instead we discover pk-fk relationship between columns of different tables using the steps outlined in Appendix A.2.

### 3.1.2 SQL QUERY GENERATION AND EXECUTION

This step of the process generates a set of SQL queries per dataset and executes it to collect the query cardinalities, the training labels. We use the workload generator proposed by (6) [2]. The generator takes as input the calculated statistics for each dataset and also a configuration that specifies the SQL

---

[2]https://github.com/DataManagementLab/zero-shot-cost-estimation, license: Apache License 2.0

Table 2: *(left)* Query Graph (heterogeneous) features. Features can be numeric ($_{\in \mathbb{R}}$) or categorical ($_{\in \mathbb{Z}}$). Symbol ♦ denote features utilized in query workload generation, and ♥ denotes ones used by our models (§5). Notes: [i] §3.1.1; [ii] Always = "Pearson"; [iii] $\in$ {join, scan}; [iv] $\in$ {=, >, <, ≥, ≤, IS, IS NOT, BETWEEN, AND}. *(right)* Query Graph depiction with corresponding SQL query statement.

| Node Type | Feature |
|---|---|
| tables | rows$^{♦♥}_{\in \mathbb{R}}$ |
| | name$^{♦}_{\in \mathbb{Z}}$ |
| correlations | correlation$^{♥}_{\in \mathbb{R}}$ |
| | validity$^{♥}_{\in \mathbb{Z}}$ |
| | type$^{[ii]}_{\in \mathbb{Z}}$ |
| ops | operator$^{♥\,[iii]}_{\in \mathbb{Z}}$ |
| predicates | predicate_operator$^{♥\,[iv]}_{\in \mathbb{Z}}$ |
| | estimated_selectivity$^{♥}_{\in \mathbb{R}}$ |
| | offset$^{♥}_{\in \mathbb{R}}$ |
| | constant$_{\in \mathbb{Z}}$ |
| | encoded_constant$_{\in \mathbb{R}}$ |

| Node Type | Feature |
|---|---|
| attributes | name$^{♦}_{\in \mathbb{Z}}$ |
| | data_type$^{♥}_{\in \mathbb{Z}}$ |
| | null_frac$^{♦♥}_{\in \mathbb{R}}$ |
| | num_unique$^{♥}_{\in \mathbb{R}}$ |
| | percentiles_100_numeric$^{♦♥}_{\in \mathbb{R}}$ |
| | percentiles_100_string$^{♦}_{\in \mathbb{Z}}$ |
| | min_string$^{♦}_{\in \mathbb{Z}}$ |
| | max_string$^{♦}_{\in \mathbb{Z}}$ |
| | min_numeric$^{♦}_{\in \mathbb{Z}}$ |
| | max_numeric$^{♦}_{\in \mathbb{Z}}$ |

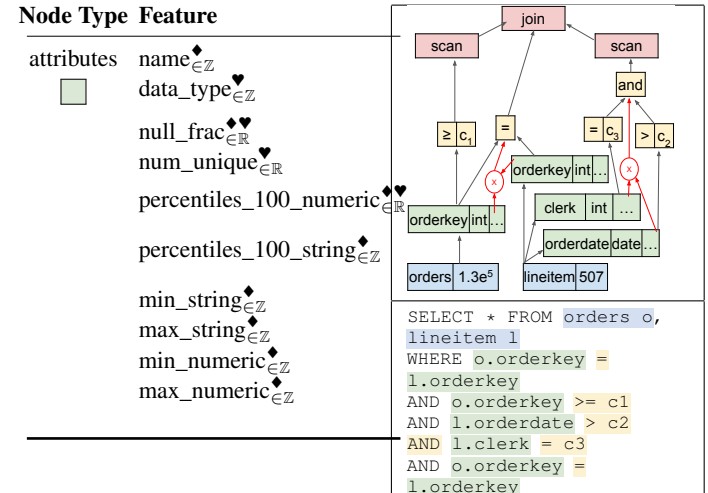

query shapes to be produced. We have modified the original workload generator so that all queries return the cardinality as their final result.

### 3.1.3 ANNOTATED QUERY GRAPH GENERATION

After the queries are executed, we convert the SQL queries into a graph representation which is annotated with statistics. We focus on transferable features and not dataset or workload specific features such as table and attribute names. This approach enables learning of zero-shot CE models. An example is shown in Table 2, on the right we present the sql query and its representation as a graph. The graph nodes are annotated with the features listed on the left of Table 2. The query graph has table nodes (in blue), column nodes (in green), operator nodes (in red), predicates (in yellow) and correlations across column of the same table (white circles with red outline). The only feature that is calculated as part of this step is the predicate *estimated_selectivity*, as both dataset statistics and query are needed. Selectivity is a fraction that represents the part of the input that satisfies a predicate (17) and is used to estimate cardinality. We estimate selectivity for filter predicate nodes using the methods described in (18).

## 4 CARDINALITY ESTIMATION MODELS

In this section, we discuss details of our model for which supports zero-shot CE.

### 4.1 DATA ENGINEERING

**Features**: The set of features (*numerical* or *categorical*) that the models use are specified in Table 2. In order to train a zero-shot model that can generalize to new datasets, we only include transferable features that are dataset-agnostic. This means that we exclude dataset-specific information such as table or column names. Furthermore, we select detailed statistics (e.g., histogram, correlation) as input features so the model can learn the underlying data distribution.

**Pre-processing**: For numerical features and the cardinality label, we employ a data transformation process to enhance their representation. First, we apply a logarithmic transformation to the data, which helps to reduce the impact of outliers. We then perform standardization to ensure that all features are on the same scale. For categorical features, we adopt one-hot encoding to allow effective learning of relationships between different categories. Additionally, we filter out queries with zero cardinality or zero predicates. In practice, such queries are relatively rare, and their inclusion could introduce noise into the training process.

## 4.2 GRAPH NEURAL NETWORK MODEL

The graph representation of SQL queries provides a natural way to capture the relationship between different elements of a query. This representation allows us to directly leverage Graph Neural Network (GNN) which is designed to learn from graph-structured data. We build a GNN model to predict cardinality of queries based on the design of the GNN-based cost model from (6). The GNN model first initializes the hidden state for each node by applying a node-type specific Multi-Layer-Perceptrons (MLP) to the node's input feature vector (a concatenation of the input feature values of that node). Since query graphs are directed acyclic graphs, the GNN model propagates information from leaf to root nodes via bottom-up message passing. This topological information flow also follows the natural query processing order. During message passing, at each node, the hidden states of its children nodes are summed up and concatenated with the original hidden state of that node, the concatenated embedding then runs through a separate node-type specific MLP to derive the updated hidden state. Finally after all nodes are updated in topological order, the hidden state of the root node is read out as the learned graph-level embedding. This embedding is passed into an MLP-based cardinality model to get the final prediction.

## 4.3 GRAPH TRANSFORMER MODEL

In addition to GNN, Graph transformer is an emerging type of neural network architecture specifically designed for graphs. Similar to traditional transformers, which capture the relationship between tokens in a sequence with self-attention, graph transformer extends the attention mechanism to graphical data, where the relationships between different nodes are constrained by the graph structure (25). Compared to traditional transformers, we make following adaptation[3]:

**Heterogeneous input embedding layer**: The SQL query graph is a heterogeneous graph with multiple node types that contain different features. For each node type, we use a separate input embedding layer to project node features to a fixed-dimension input node embedding.

**Shortest distance spatial encoding**: Following the methodology of Graphormer (24), we compute the shortest path distance between node pairs to construct a spatial encoding matrix. This matrix is incorporated as a learnable bias that adds to the input before the softmax in the self-attention block.

**Directional causal mask**: To ensure information flow adheres strictly to the DAG's structure, we implement a causal masking mechanism based on the topological ordering of the nodes.

**Virtual node readout**: Inspired by Graphormer, we augment the query graph by introducing a virtual node as the direct child of all other nodes. Given that the virtual node attends to all other nodes in the query graph, its embedding serves as the graph-level representation for cardinality prediction.

Figure 2 shows an overview of the graph transformer cardinality prediction model. The backbone of the model architecture is derived from encoder-only transformers. The node feature encoder embed the node features to vectors and pass those vectors to $N$ transformer blocks with modified self-attention layers using additional causal mask and spatial encoding.

## 5 EXPERIMENTAL FINDINGS

In this section, we present our empirical findings, evaluating the accuracy of various model configurations for cardinality prediction. We focus on the *Single Table* and *Binary Join* CardBench datasets that contain queries with 0 to 1 joins, as often the majority of queries fall in this category (in the recently released dataset by Amazon (20) more than 90% of the queries have 0 to 1 joins).

### 5.1 EVALUATION SETUP

#### 5.1.1 MODEL CONFIGURATIONS

We assess cardinality prediction performance using three distinct ML model configurations: instance-based, zero-shot, and fine-tuned.

---

[3]Figure 2 in the appendix illustrates the architecture of the attention block of the graph transformer model designed to embed SQL query graphs.

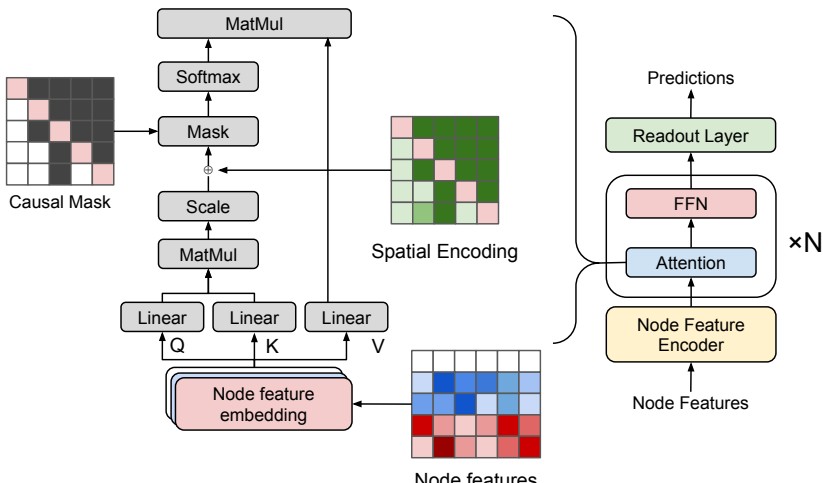

Figure 2: An overview of the graph transformer.

**Instance-based Models:** In this configuration, we train and evaluate individual models using a single dataset. For each dataset, queries are randomly partitioned into training and validation sets with 85:15 train-validation split. An additional 500 queries are reserved as a standalone test set to evaluate final model accuracy.

**Zero-shot Models:** This configuration investigates the generalizability of cardinality prediction models to out-of-distribution data, i.e., data from another dataset apart from the training data. We train and validate the model on queries from 19 datasets, maintaining the 85:15 train-validation split. The model is then tested on the standalone test set of the remaining 20th dataset. This test set contains the same queries used in the instance-based setting to ensure a fair comparison.

**Fine-tuned Models:** This configuration investigates whether fine-tuning a pre-trained zero-shot model improves accuracy and sample efficiency. We fine-tune the model initially trained on 19 datasets using the 20th dataset. The fine-tuned model's accuracy is then evaluated using the same 500 holdout queries from this 20th dataset.

The average training times under each configuration are[4] 1.3hr, 1.4hr and 11min (single-table queries), and 1.3hr, 1.5hr, and 11min (binary join queries) for the GNN-based model on a 8-core VM (4GB memory). For the Transformer-based model the average training times are 3.3hr, 11.8hr and 1.6hr (single-table queries), 11.1hr, 11.1hr and 2.1hr (binary join queries) on a V100 GPU. We observe that training the zero-shot model over extended epochs can often result in lower accuracy due to over-fitting to the training set. Therefore, we cap the maximum number of training epochs for zero-shot training to 20, and set a maximum training epochs of 100 for the other two configurations.

The inference time for the GNN model is 35ms and for the transformer model is 97ms. The inference time of the zero-shot, fine-tuned and instance-based configurations of the same model type are the same since they share the same architecture and model size. The graph transformer model size is 33.6MB, with 8.4M parameters in total, the GNN model size is 7.5MB, with 1.88M parameters in total.

### 5.1.2 BASELINE

We compare the learned cardinality approaches against a the cardinality estimation of PostgreSQL[5]. To get the estimates we load the CardBench data in PostgreSQL and use the *explain* command, similar to (6). We call this baseline *Postgres*.

---

[4]Listed in instance-based, zero-shot, fine-tuned order.
[5]https://www.postgresql.org/

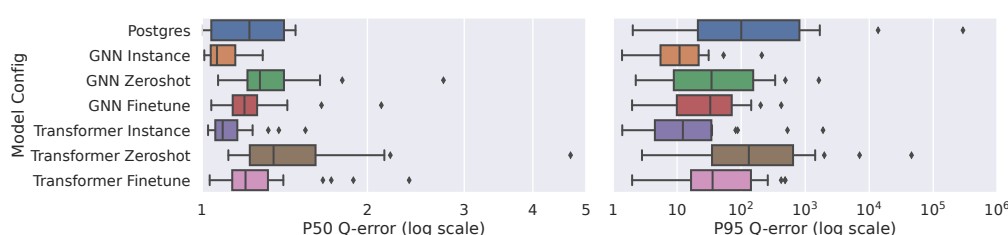

Figure 3: Box plots of P50 (left) and P95 (right) q-errors of different model configurations for queries on a single table, aggregated across 20 test datasets.

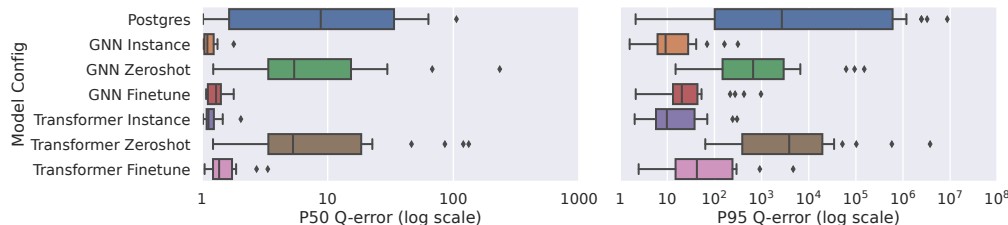

Figure 4: Box plots of P50 (left) and P95 (right) q-errors of different model configurations for queries contains a binary join, aggregated across 20 test datasets.

## 5.2 ACCURACY OF CARDINALITY PREDICTION

We evaluate the accuracy of cardinality prediction by comparing the baseline method, GNN models, and graph transformer models, each under different configurations as described in Section 5.1.1. We run a single experiment on each of the 20 test datasets in CardBench per model configuration. We then collect and aggregate the accuracy metrics across all experiments per model configuration. For training instance-based and zero-shot models, 4500 queries are randomly selected per dataset to form the training and validation set. Fine-tuning utilizes a smaller subset of 500 queries per dataset.

### 5.2.1 SINGLE TABLE CARDINALITY PREDICTION

Figure 3 shows the box-and-whisker plots of P50 and P95 q-errors of different model configurations for queries on a single table, aggregated across 20 test datasets in CardBench. The baseline algorithm demonstrates strong median accuracy results in single-table cardinality prediction, with the majority of datasets achieving a q-error below $1.5$. However, accuracy degrades significantly at the tail due to the failure of both the uniform distribution and independence assumptions that the baseline approach makes, especially for more complex queries.

GNN-based models outperform transformer-based models in most cases, despite the latter's larger receptive field through attention mechanisms. This suggests that iterative message passing along query graph paths adequately captures the structure of SQL queries. Notably, instance-based models achieve the highest accuracy, with the GNN model demonstrating an average P50 q-error of $1.1$ and a P95 q-error of $24.08$, and the transformer model demonstrating an average P50 q-error of $1.15$ and a P95 q-error of $136.87$. It is achieved by implicitly learning dataset-specific joint-column data distributions, enabling tailored predictions. In contrast to instance-based models, zero-shot models struggle to generalize such knowledge to unseen datasets, resulting in a lower accuracy (especially at

Table 3: Average Q-Error percentiles, from results presented in Figure 4. The best accuracy (excluding instance-based models) are in bold.

| Model | $Q_{err}^{50}$ | $Q_{err}^{75}$ | $Q_{err}^{90}$ | $Q_{err}^{95}$ | $Q_{err}^{50}$ | $Q_{err}^{75}$ | $Q_{err}^{90}$ | $Q_{err}^{95}$ |
|---|---|---|---|---|---|---|---|---|
| | | no join | | | | binary join | | |
| Postgres | **1.03** | 18.01 | 7864.88 | 12296.35 | 5.29 | 21.03 | 38557.10 | 411511.45 |
| GNN Instance | 1.10 | 1.64 | 6.00 | 24.08 | 1.17 | 1.82 | 10.92 | 37.55 |
| GNN Zeroshot | 1.39 | 11.52 | 51.09 | 171.25 | 22.66 | 102.12 | 3430.23 | 16271.84 |
| GNN Finetune | 1.22 | **2.00** | **8.07** | **41.50** | 1.32 | **2.45** | **13.62** | **109.77** |
| Transformer Instance | 1.14 | 1.86 | 9.46 | 140.67 | 1.20 | 1.99 | 11.93 | 43.96 |
| Transformer Zeroshot | 1.60 | 8.38 | 666.31 | 2954.98 | 24.88 | 264.06 | 6000.99 | 228499.79 |
| Transformer Finetune | 1.25 | 2.81 | 13.80 | 79.52 | 1.57 | 4.19 | 47.14 | 355.57 |

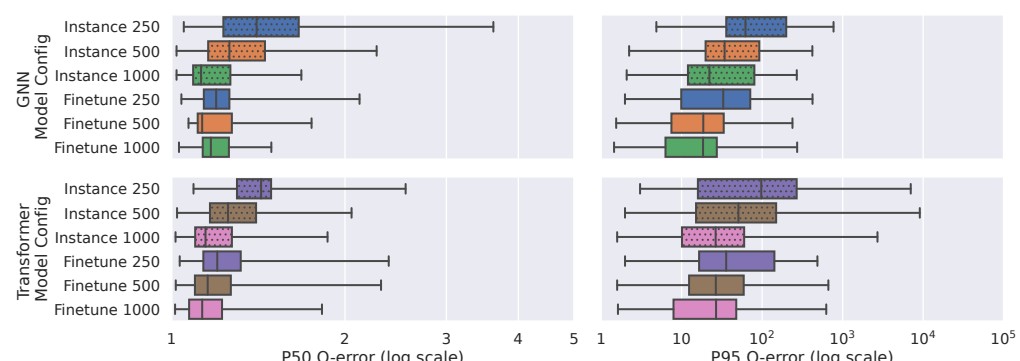

Figure 5: Box plots of P50 (left) and P95 (right) q-errors of instance-based vs. fine-tuned models of GNN (top) and Transformer (bottom) with varying training sample size for queries on a single table, aggregated across 20 test datasets.

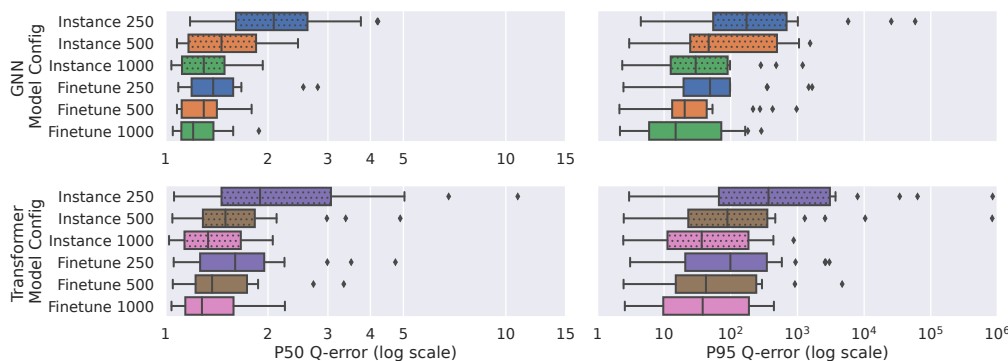

Figure 6: Box plots of P50 (left) and P95 (right) q-errors of instance-based vs. fine-tuned models of GNN (top) and Transformer (bottom) with varying training sample size for queries contains a binary join, aggregated across 20 test datasets.

the tail). Fine-tuning pre-trained models, even with a modest 500 samples, significantly improves accuracy for both GNN and transformer models, highlighting the importance of incorporating table and column-specific distributions for accurate cardinality prediction. Table 3 further breaks down the statistics shown in Figure 3.

### 5.2.2 BINARY JOIN CARDINALITY PREDICTION

Cardinality prediction for queries with binary joins is much more challenging than single table because cross-table distribution and multi-column correlation, which are difficult to model, significantly impact join cardinality. Figure 4 shows the P50 and P95 q-errors for queries with binary joins. In contrast to single-table prediction, the baseline algorithm exhibits significant inaccuracy in binary join query CE, with a median q-error of $55.54$ and P95 q-error of $4.4 \times 10^6$, respectively. This appears to be a result of the oversimplified assumption that queries join two tables via single-column primary and foreign keys, without considering more complex cross-table relationships that are intractable using traditional heuristics.

Learning-based models significantly outperform the baseline though, particularly at the 95th percentile. Instance-based models, which learn cross-table distributions within a dataset, achieve low median q-errors of 1.16 (GNN) and 1.20 (transformer). While their P95 q-errors are higher (37.55 for GNN, 43.96 for transformer), the results suggest that estimating join cardinality via learned distributions is feasible. However, out-of-distribution cardinality prediction for binary joins proves more challenging, as zero-shot models experience a dramatic increase in q-errors (up to 20x at the median, 5300x at P95). As with single-table results, even a small amount of data used for fine-tuning improves the accuracy of pre-trained models considerably. Table 3 further breaks down the statistics shown in Figure 4.

## 5.3 VARYING SAMPLE SIZE

We study the impact of varying amounts of training data on model accuracy by considering three sample sizes: 250, 500, and 1000. For each sample size, we train an instance-based model or fine-tune a pre-trained model, and evaluate it on the same 500 samples as described in Section 5.1.1. We repeat this process for both the GNN and transformer-based models. The results for single table and binary join cardinality predictions are summarized in figure 5 and figure 6 respectively. We observe same patterns between GNN and transformer. The model accuracy improves as more training data is provided, for both instance-based and fine-tuned models.

Comparing instance-based training with fine-tuning, we find that the fine-tuned models are more sample-efficient. Fine-tuned models achieve similar median and tail Q-errors as instance-based models trained with twice the amount of data. This is evidence that fine-tuned models benefit from the generalization power of a pre-trained model, which has learned from a large number of data patterns over many datasets. When using only 500 samples from each of the binary join datasets, the average P50 and P95 Q-errors across 20 datasets are: 1.57 & 280 for the instanced-based GNN model, and 1.32 & 120 for the fine-tuned GNN model. There is a significant improvement especially for the tail queries due to learnings transferred from more diverse data patterns seen by the pre-trained model. Nonetheless, the accuracy gap between instance-based and fine-tuned models reduces as the sample size increases. This suggests that fine-tuning on a pre-trained model is more advantageous when there is limited amount of training data available for a new dataset.

## 6 CONCLUSION AND BROADER IMPACT

In this paper, we release CardBench a benchmark for CE in relational databases. CardBench contains code to 1-) compute various data statistics, 2-) generate queries and execute them, and 3-) create the training data (the annotated querygraphs). We are also releasing three training datasets with queries over 20 datasets and their true cardinalities: one containing queries with single table queries, one with binary joins, and the third with multi-join queries. We trained both instance-based and zero-shot models, and observed that CE is a extremely challenging problem to learn in a zero-shot manner. However, when using pre-trained models with a small amount of fine-tuning, we can achieve accuracy similar to instance-based models, with much lower training overhead.

With CardBench, we hope to provide a systematic mechanism to track progress, especially on recent directions such as pre-trained CE models. Moreover, we think that CardBench is highly important to foster further research on CE models, from the ML and DB communities. The training datasets were constructed with high overhead (14 cpu years of query runtime). Releasing such a training set lowers the barrier of entry for developing and training new CE models. Furthermore, we hope that the benchmark itself will be extended with new datasets and more complex queries using the tools and code we provide and thus push the boundaries of learned CE models further.

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

# A APPENDIX

## A.1 WORKLOAD STATISTICS

Table 4: Query Cardinalities. (The Multi-join Cardinalities are being calculated and will be added soon.)

| Database Name | Single Table | | | Binary Join | | | Multi-Join | | |
|---|---|---|---|---|---|---|---|---|---|
| | AVG | MAX | STDDEV | AVG | MAX | STDDEV | AVG | MAX | STDDEV |
| airline | 196.8K | 11.4M | 1.3M | 4.3M | 4.3M | 4.3M | | | |
| sample_covid19_weathersource_com | 225.6M | 2.8B | 656.4M | 4.3B | 4.3B | 4.3B | | | |
| sample_idc_v10 | 4.1M | 29.1M | 9.7M | 5.1M | 5.1M | 5.1M | | | |
| sample_usfs_fia | 2.3M | 30.1M | 6.2M | 8.2M | 8.2M | 8.2M | | | |
| sample_cms_synthetic_patient_data_omop | 40.8M | 748.9M | 115.6M | 304.1M | 304.1M | 304.1M | | | |
| sample_crypto_bitcoin_cash | 36.8M | 146.0M | 53.8M | 56.2M | 56.2M | 56.2M | | | |
| sample_human_variant_annotation | 4.2M | 79.2M | 12.5M | 1.8M | 1.8M | 1.8M | | | |
| sample_wikipedia | 162.3M | 1.1B | 357.5M | 16.1B | 16.1B | 16.1B | | | |
| tpch_10G | 2.6M | 60.0M | 9.0M | 110.6M | 110.6M | 110.6M | | | |
| sample_samples | 17.1M | 188.3M | 32.2M | 15.6M | 15.6M | 15.6M | | | |
| consumer | 1.4M | 12.1M | 3.2M | 2.2M | 2.2M | 2.2M | | | |
| sample_open_targets_genetics | 10.4M | 222.0M | 32.0M | 10.2B | 10.2B | 10.2B | | | |
| accidents | 200.1K | 954.0K | 318.6K | 382.1K | 382.1K | 382.1K | | | |
| sample_stackoverflow | 9.1M | 141.9M | 21.3M | 11.7M | 11.7M | 11.7M | | | |
| employee | 883.2K | 8.5M | 1.7M | 1.4M | 1.4M | 1.4M | | | |
| movielens | 687.5K | 8.5M | 1.6M | 1.5M | 1.5M | 1.5M | | | |
| sample_ethereum_blockchain | 19.6M | 181.8M | 40.4M | 27.0M | 27.0M | 27.0M | | | |
| sample_github_repos | 24.9M | 230.9M | 62.2M | 466.7M | 466.7M | 466.7M | | | |
| sample_geo_openstreetmap | 27.2M | 560.7M | 75.4M | 19.9M | 19.9M | 19.9M | | | |
| sample_uspto_oce_claims | 35.8M | 82.0M | 37.6M | 45.2M | 45.2M | 45.2M | | | |

## A.2 PRIMARY-FOREIGN KEY RELATIONSHIPS

Primary foreign key (pk-fk) relationships are part of a database schema design (17). A pk-fk relationship is a logical connection between the columns of two tables semantically linking their information. In the real world, most of the times two tables are joined using the columns that participate in the pk-fk relationship. Naturally we want to use pk-fk relationships when creating synthetic workloads. A challenge we faced is that open source datasets do not usually specify pk-fk

relationships, instead we discover pk-fk relationship between columns of different tables using the steps outlined below. One assumptions we make is that data is organized in a snowflake schema, this is the norm in practice and what is assumed by most database benchmarks (TPC-H (3), SSB (16), TPC-DS(2), TPC-C(1), JOB(10)).

1. Identify candidate primary keys per table using a uniqueness threshold and choose one with the most unique values. A primary key should be unique for each row of a table (17), we relax this condition and require that for a column to be a primary key more than 90% of rows must be unique values.

2. Split tables of a dataset in fact and dimension tables[6]. The criteria we use is a size threshold (fact tables should be a few times larger than dimension tables) as well as a check if a primary key exists for dimension tables.

3. For each dimension table and pk column identify a fk column in the fact table(s). Out of all possible candidates we choose the pk-fk pair with the most join results, which we obtain by running a join query using the candidate pk-fk column pair as the join condition.

---

[6]https://en.wikipedia.org/wiki/Snowflake_schema

