# OpenReview forum: "CardBench: A Benchmark for Learned Cardinality  Estimation in Relational Databases"
_ICLR.cc/2025/Conference — Submitted to ICLR 2025_

### Official Review · Reviewer_uy6z · 2024-11-04

**Soundness:** 2
**Presentation:** 2
**Contribution:** 2
**Rating:** 5
**Confidence:** 4

**Summary:**

The paper presents CardBench, a benchmark for evaluating learned cardinality estimation models in relational databases. CE is vital for optimizing query performance, yet traditional methods often lack accuracy. CardBench offers a diverse set of datasets and queries, featuring hundreds of thousands of queries across 20 real-world datasets, enabling systematic assessments of new CE approaches.
Experiments with GNN and transformer-based models were conducted under three setups: instance-based, zero-shot, and fine-tuning. While zero-shot estimation shows promise for simple queries, its accuracy declines with complex joins. However, fine-tuning pre-trained models can enhance performance significantly, reducing training costs.
The authors emphasize that CardBench will lower barriers for research in learned CE and encourage further exploration from the machine learning community.

**Strengths:**

- CardBench includes a broad range of datasets, 20 distinct real-world datasets, that provide a more comprehensive evaluation framework for learned cardinality estimation models compared to existing benchmarks.
- The benchmark facilitates systematic testing of various learned CE approaches, allowing researchers to assess model performance comprehensively. By open-sourcing the benchmark, query generator, and associated scripts, the authors foster collaboration and further research in the field, lowering barriers for other researchers to experiment with learned cardinality estimation.
- The paper is clearly written and organised.

**Weaknesses:**

- There are several typos that need to be corrected in future revisions.
- It would be beneficial to include more detailed statistical experimental results. While boxplots effectively illustrate the distribution of results, providing exact values enables precise comparisons between data points.
- The descriptions of the GNN and Transformer models used in the experiments are lacking. More detailed explanations would help readers gain better insights into the methodologies.
- The paper should discuss how the performance of other state-of-the-art methods compares on CardBench, which would provide context and strengthen the findings.

**Questions:**

Please see weaknesses.

---

> ### Author Response · Authors · 2024-11-19
>
> Thank you for your valuable feedback, please find below response to some of the points raised in your review.
>
> W1: Thank you for pointing out typos, we will correct them.
>
> W2: Thank you for the suggestion. We added Table 3 with detailed statistical experimental results for the results shown with box plots in Figure 3 and 4 (Q90 and Q99).
>
> W3: We have added more details about the GNN and transformer models we use in the Appendix of the revised manuscript we submitted to open review. Do you have any specific suggestions for additional details we can add?
>
> W4: CardBench, which is the focus of the paper, can be used to train, evaluate and compare pre-trained, zero-shot methods (which are the focus of CardBench), workload driven and instance based methods.
> Regarding the models proposed, their use in this paper is to showcase that a zero-shot/ pre-trained approach is possible for cardinality estimation. We think a comparison with other approaches (non zero-shot) is out of the scope of this paper ([1] has already made this case). Nevertheless, we added a comparison with the cardinality estimator of PostgreSQL as a baseline to represent a state-of-the-art non-learned approach.  The cardinality estimator of PostgreSQL is used as the baseline in most relevant papers [1, 2, 3, 4]. We have added two revised figures in an open review revision of the paper (cf Figure 3 and 4). Our methods outperform the traditional no training methods (e.g. summary based methods) that PostgreSQL uses and more importantly they are more robust at the tail.
>
>
>
>
> References
> [1] Benjamin Hilprecht and Carsten Binnig. 2022. Zero-shot cost models for out-of-the-box learned cost prediction. Proc. VLDB Endow. 15, 11 (July 2022), 2361–2374. https://doi.org/10.14778/3551793.3551799
>
> [2] Zongheng Yang, Wei-Lin Chiang, Sifei Luan, Gautam Mittal, Michael Luo, and Ion Stoica. 2022. Balsa: Learning a Query Optimizer Without Expert Demonstrations. In Proceedings of the 2022 International Conference on Management of Data (SIGMOD '22). Association for Computing Machinery, New York, NY, USA, 931–944. https://doi.org/10.1145/3514221.3517885
>
> [3] Ryan Marcus, Parimarjan Negi, Hongzi Mao, Chi Zhang, Mohammad Alizadeh, Tim Kraska, Olga Papaemmanouil, and Nesime Tatbul. 2019. Neo: a learned query optimizer. Proc. VLDB Endow. 12, 11 (July 2019), 1705–1718. https://doi.org/10.14778/3342263.3342644
>
> [4] Kipf, Andreas, et al. "Learned cardinalities: Estimating correlated joins with deep learning." arXiv preprint arXiv:1809.00677 (2018).

---

### Official Review · Reviewer_EaZ6 · 2024-11-04

**Soundness:** 3
**Presentation:** 2
**Contribution:** 3
**Rating:** 5
**Confidence:** 3

**Summary:**

This paper proposes the cardbench, a benchmark dataset for cardinality estimation in relational databases, which contains over 20 distinct real-world databases. This proposed dataset shows clear diverse and proposes several baseline models including the GNN and transformer modules to form a benchmark evaluation of this benchmark dataset.

**Strengths:**

+ A large-scale benchmark dataset for Cardinality Estimation
+ Good evaluations and benchmark analyses.
+ Clear organizations and easy to read.

**Weaknesses:**

+ My major concern is about the benchmark contributions to this research field. In the related work section, the authors talked about how existing benchmarks only contain one or two datasets which is insufficient for testing pretraining models. Thus my question is, how does this proposed benchmark test these pretrained models? It seems the authors only test typical GNN or Transformer architectures.
This dataset seems not to be comprehensive as a benchmark in this research field.
+ Besides, gathering different datasets or re-organizing them also may help the aforementioned problems. Why is this proposed benchmark unique? I am concerned about the realistic usage and if this is a real problem for the community.
+ There are also several minor issues, including the presentations for the Sect. 3 and the table format for table 1. These presentations could be improved.

**Questions:**

Please refer to the weakness section and I am concerned about the realistic usage of this proposed benchmark or it is an ``made" problem.

---

> ### Author Response · Authors · 2024-11-19
>
> Thank you for your valuable feedback, please find below response to some of the points raised in your review.
>
> W1,W2: Existing benchmarks and their training data are insufficient to train zero-shot and GNN models. [1] showed that for database tasks at least 20 datasets, which is supported by our experiments as well, are needed to train a zero-shot model. Thus existing datasets with labeled data/benchmarks that include 1-3 datasets are not sufficient to be used to train models that are going to be used in pre-trained or zero-shot approach. Data sourcing, preparation and collection of training labels for database tasks, which require running thousands of queries, are time-consuming and big barriers for the creation of models.
> CardBench is a collection of datasets, collected and calculated statistics about these datasets, queries along with training labels (cardinalities) and training data (queries represented as graphs annotated with statistics). The combination of all these assets make CardBench much more than a collection of datasets. Additionally, CardBench includes scripts to extend the benchmark with more datasets and queries if needed.
> We believe the CardBench data and queries can enable research on zero-shot/pre-trained models for cardinality estimation by making the creation of models easier and also by allowing easy model comparison.
> Further CardBench can test zero-shot/pre-trained models on 20 datasets, using a leave-one-out setting to stress a model’s generalization ability. This is a more realistic use case compared to evaluating on 1-3 datasets that other benchmarks include.
> As for the model architectures please see our general rebuttal response.
>
> W3: We will work on the presentation and improve the parts mentioned.
>
> [1] Benjamin Hilprecht and Carsten Binnig. 2022. Zero-shot cost models for out-of-the-box learned cost prediction. Proc. VLDB Endow. 15, 11 (July 2022), 2361–2374. https://doi.org/10.14778/3551793.3551799

---

### Official Review · Reviewer_4NBw · 2024-11-09

**Soundness:** 2
**Presentation:** 2
**Contribution:** 2
**Rating:** 3
**Confidence:** 3

**Summary:**

The paper introduces CardBench, a benchmark for evaluating learned cardinality estimation models in relational databases. Cardinality estimation is vital for query optimization, but existing models lack a comprehensive benchmark for systematic evaluation. CardBench provides this with extensive queries across 20 real-world databases. The study evaluates GNN and transformer-based models in instance-based, zero-shot, and fine-tuned setups, highlighting challenges in zero-shot accuracy with complex queries but demonstrating the potential of fine-tuning pre-trained models. By releasing scripts and data, the authors encourage further research, showing that pre-trained models can achieve high accuracy with reduced training overhead.

**Strengths:**

S1: Cardinality estimation is critical for the database community.

S2: This paper proposes several datasets, which will be useful for academic and industry communities.

S3: Zero-shot cardinality estimation seems to be useful.

**Weaknesses:**

W1: The details of the datasets are unclear.

W2: The baselines are too weak and lack state-of-the-art and representative cardinality estimation baselines.

W3: The experiments lack detailed analysis of the proposed models regarding the zero-shot setting.

**Questions:**

D1: This paper proposes several synthetic cardinality estimation datasets. But what are the patterns and distributions in these datasets?

D2: Why are these datasets comprehensive and diverse? From a database view, are these proposed datasets complete? It is better to add more justifications.

D3: This paper proposes zero-shot estimation but only investigates limited GNNs. In cardinality estimation, whether sampling-based approaches or summary-based approaches, all work under a zero-shot setting. These approaches do not need any training instances. But this paper does not explore these approaches.

D4: Even for GNN-based baselines, they are too simple.

D5: How about the training time and inference time compared with the time of the training-free methods? If the training-free methods are better than the proposed method, it is meaningless to propose a pre-trained model.

D6: This paper only explores the performance on cardinality estimation error. But how to prove it can achieve better performance on downstream applications? It is better to deploy this model on a real database system to see its improvements.

---

> ### Author Response · Authors · 2024-11-19
>
> Thank you for your valuable feedback, please find below response to some of the points raised in your review.
>
> D1, D2: Please see our answer to point (B) in the response to all reviewers. If you have suggestions for additional statistics we would be happy to extend Table 1 and 4.
> The 3 training sets we created (Single Table, Binary Join, Multi-Join) progressively increase the difficulty of the prediction task and are modeled after common patterns seen in real database products (In [1] Amazon reports the majority of production database queries have 0-2 joins).
>
> D3, D5: We have updated the paper to include training and inference times. The GNN model has an inference time of 35ms, while the transformer model takes 97ms. Furthermore, we now compare our approach to PostgreSQL's cardinality estimation (Figures 3, 4). PostgreSQL, a widely used database, serves as a baseline in most relevant papers [2-5]. Our methods outperform its traditional summary-based techniques and show greater robustness, especially at the tail.
> In general, the research on instance-based learned cardinality estimation has shown they outperform training-free (traditional) methods [2-4, 8-16].  The cost to train and the investment needed to build training pipelines has been a major blocker for wide adoption of such techniques despite their high accuracy. Thus if pre-trained or zero-shot models prove to be equally as performant as instance-based methods then we could use them in practice.
>
>
> D3, D4: The scientific problem we targeted in this paper is if cardinality estimation can be done in a zero-shot / pre-trained way. A positive answer motivates a new benchmark that will allow the training and testing of such models. Using GNN and transformer models, we show that the problem is not zero-shot but needs fine-tuning. Potentially there are other model architectures that could work better for this problem. The contribution of this paper is showing that cardinality estimation can be solved with pre-trained models, coupled with a smaller number of training data used for fine-tuning and introduce a new benchmark that can enable research in this area. We hope to motivate the research community to focus on zero-shot/pre-trained models for cardinality estimation and use CardBench to do so.
>
> D6: Q-error has become the standard metric for evaluating cardinality estimation [2-5, 8-16]. Deploying models in a database system can indeed provide useful insights but requires query execution performance analysis to correctly attribute performance wins or shortcomings.
> It is well accepted in the database community that accurate cardinality estimation is beneficial [6]. Cardinality estimation is a major building block in many database problems, including all recommenders (index, materialized views, partitioning), query optimization, as well as workload management and scheduling. As such accurate  cardinality estimation is critical for high-performance databases.
>
> [1] Alexander van Renen et al 2024. Why TPC is Not Enough: An Analysis of the Amazon Redshift Fleet VLDB 2024
>
> [2] Zongheng Yang et al. Balsa: Learning a Query Optimizer Without Expert Demonstrations SIGMOD '22
>
> [3] Ryan Marcus, et al VLD 2019. Neo: a learned query optimizer.
>
>
> [4] Kipf, Andreas, et al. "Learned cardinalities: Estimating correlated joins with deep learning." arXiV 2018
>
> [5] Benjamin Hilprecht and Carsten Binnig. VLDB 2022. Zero-shot cost models for out-of-the-box learned cost prediction.
>
>
> [6]: Leis, Viktor, et al. "How good are query optimizers, really?." Proceedings of the VLDB Endowment 9.3 (2015): 204-215.
>
> [7] Unnesting Arbitrary Queries Thomas Neumann and Alfons Kemper https://cs.emis.de/LNI/Proceedings/Proceedings241/383.pdf
>
> [8]: Benjamin Hilprecht, et al DeepDB: learn from data, not from queries! Proc. VLDB 2020
>
> [9]: Getoor, Lise, Benjamin Taskar, and Daphne Koller. "Selectivity estimation using probabilistic models." SIGMOD 2001.
>
> [10]: Liu, Henry, et al. "Cardinality estimation using neural networks." Proceedings of the 25th Annual International Conference on Computer Science and Software Engineering. 2015.
>
> [11]: Dutt, Anshuman, et al. "Selectivity estimation for range predicates using lightweight models." Proceedings of the VLDB Endowment 12.9 (2019): 1044-1057.
>
> [12] Kipf, Andreas, et al. "Learned cardinalities: Estimating correlated joins with deep learning." arXiv preprint arXiv:1809.00677 (2018).
>
> [13] Sun, Ji, and Guoliang Li. "An end-to-end learning-based cost estimator." arXiv preprint arXiv:1906.02560 (2019).
>
> [14] Woltmann, Lucas, et al. "Cardinality estimation with local deep learning models." Proceedings of the second international workshop on exploiting artificial intelligence techniques for data management. 2019.
>
> [15] Parimarjan Negi et al Robust Query Driven Cardinality Estimation under Changing Workloads. VLDB 2016
>
>
> [16] Negi, Parimarjan, et al. "Flow-loss: Learning cardinality estimates that matter." arXiv preprint arXiv:2101.04964 (2021).

---

### Official Review · Reviewer_vebN · 2024-11-09

**Soundness:** 2
**Presentation:** 2
**Contribution:** 2
**Rating:** 5
**Confidence:** 3

**Summary:**

This paper proposes a benchmarking framework aimed at cardinality estimation within relational databases, using advanced learning methods like GNNs and Transformers.

**Strengths:**

The paper details a systematic data preparation process, including SQL query generation, dataset statistics calculation, and annotated query graph creation, which offers a replicable approach for dataset-agnostic testing.

The paper’s emphasis on creating a model that can generalize to unseen datasets is a unique and relevant shift in the CE field, considering the increasing need for adaptable models in dynamic data environments.

**Weaknesses:**

1. The authors said that they have collected data from 20 datasets with diverse sources compared with existing benchmarks. But I can't see this comparsion to conclude that how novel this part is. It deserves detailed discussion.

2. Although it includes single-table and multi-table queries, it lacks support for deeply nested and highly complex SQL queries, which are common in real-world database applications. This limitation in query complexity could lead to suboptimal model performance in practical scenarios.

3. Only q-error is used as the main evaluation metric, which may not fully capture model performance. Additional dimensions, such as runtime and resource consumption, could provide a more comprehensive assessment.

**Questions:**

see weakness above

---

> ### Author Response · Authors · 2024-11-19
>
> Thank you for your valuable feedback, please find below response to some of the points raised in your review.
>
> W1: Although zero-shot and pre-trained approaches are popular there is limited research on this  for the cardinality estimation task. The lack of benchmarks (datasets and queries) to allow the development and testing of such models is a significant barrier for entry (existing benchmarks usually include 1 to 3 datasets which are not sufficient for a zero-shot /pre-trained setting [2]). The novelty of our benchmark is the inclusion of 20 datasets, as well as different query workloads executed on those datasets with true cardinalities which constitutes training and testing data. Moreover, as we release the code, the benchmark datasets and workloads are extensible in future.
>
>
> Regarding the diversity of datasets and workloads, please see our answer to point (B) in the response to all reviewers as well as the changes in the revised paper submitted to open review.
> If you have suggestions for additional diversity statistics we would be happy to extend Table  1 and 4.
>
> W2: We are not aware of any ML-based cardinality estimation work that handles nested queries. A popular benchmark used in instance-based cardinality estimation is the JOB benchmark [9], which uses a dataset with real world data (similar to ours) uses relatively simple queries (multiple joins, multiple filters, no nesting or UDFs) and shows the inefficiency of cardinality estimators.
>
> Moreover, we are targeting the building blocks of cardinality estimation, focusing on a single query block. The query input to an SQL optimizer (a SQL query) gets transformed into a simpler query before an execution plan is generated. These simplification transformations include redundant join elimination, predicate pushdown, unnesting of nested queries, among many others. Hence, the optimizers handle nested queries either by flattening them, or by optimizing them one nested level at a time, from inner to outer. So, cardinality estimation of single blocks is a key building block, which we target in this paper. We agree that some workloads also include more complex SQL functionality (such as window functions), but database literature points to the majority of queries being simpler rather than complex, for example in [1] 1 to 2 join queries without nesting is the common case.
>
>
> W3: The research literature for cardinality estimation widely uses Q-error as the de facto metric  [2-9]. Q-error is normalized as absolute values are not as meaningful with so many different sized tables. Nevertheless, Q-error evaluates only the accuracy of the cardinality estimation. The latency of the cardinality estimation is critical as query optimization (that uses cardinality estimation) is latency sensitive. We added the inference time for each model in the paper. For the GNN model inference time is 35ms, for the transformer model it is 97ms. The inference time of the zero-shot, fine-tuned and instance-based model of the same model type are the same since they have the same architecture and model size. We also report the sizes of the models,  the graph transformer model size is 33.6MB, with 8.4M parameters in total, the GNN model size is 7.5MB, with 1.88M parameters in total. We would be happy to add more metrics if you have any suggestions.
>
>
> [1] Alexander van Renen et al 2024. Why TPC is Not Enough: An Analysis of the Amazon Redshift Fleet VLDB 2024
>
> [2] Benjamin Hilprecht and Carsten Binnig. 2022. Zero-shot cost models for out-of-the-box learned cost prediction. VLDB 2022
>
>
> [3]: Liu, Henry, et al. "Cardinality estimation using neural networks." Proceedings of the 25th Annual International Conference on Computer Science and Software Engineering. 2015.
>
> [4]: Dutt, Anshuman, et al. "Selectivity estimation for range predicates using lightweight models." Proceedings of the VLDB Endowment 12.9 (2019): 1044-1057.
>
> [5] Kipf, Andreas, et al. "Learned cardinalities: Estimating correlated joins with deep learning." arXiv preprint arXiv:1809.00677 (2018).
>
> [6] Sun, Ji, and Guoliang Li. "An end-to-end learning-based cost estimator." arXiv preprint arXiv:1906.02560 (2019).
>
> [7] Woltmann, Lucas, et al. "Cardinality estimation with local deep learning models." Proceedings of the second international workshop on exploiting artificial intelligence techniques for data management. 2019.
>
> [8] Negi, Parimarjan, et al. "Flow-loss: Learning cardinality estimates that matter." arXiv preprint arXiv:2101.04964 (2021).
>
> [9] Parimarjan Negi, Ziniu Wu, Andreas Kipf, Nesime Tatbul, Ryan Marcus, Sam Madden, Tim Kraska, and Mohammad Alizadeh. 2023. Robust Query Driven Cardinality Estimation under Changing Workloads. Proc. VLDB Endow. 16, 6 (February 2023), 1520–1533. https://doi.org/10.14778/3583140.3583164

---

> > ### Comment · Reviewer_vebN · 2024-11-27
> > **Thanks for your response**
> >
> > Thank you for your response. What is the motivation of the inclusion of 20 datasets? Does ML-based methods touch the performances limit on existing dataset?
> > Besides, I agree that this paper is more suitable for a conference in database ares based on your cited paper above, where there are more experts in the field. Thus, I will keep my score.

---

> > > ### Author Response · Authors · 2024-11-27
> > > **Thank you for your comment**
> > >
> > > Previous work that studied zero-shot/pre-trained models for database tasks ([2], Figure 12) shows that model accuracy increased as more datasets are used during training. In all cases they considered at least 15 datasets were needed to get close to each model's highest accuracy.
> > > Thus we choose to include 20 datasets and create training data for the cardinality estimation task to allow testing and training of zero-shot/pre-trained models. Existing cardinality estimation benchmarks do not include a sufficient number of datasets to train models that will be used in a zero-shot/pre-trained setting as we discuss in section 2.

---

### Author Response · Authors · 2024-11-19
**General Rebuttal Response**

We first want to thank the reviewers for the time and effort they put in the reviews. We appreciate the feedback and we will use it to improve our paper. Here, we want to reply to concerns raised in more than one review. We also reply to each of the reviews separately.

We submitted a revised paper to open review to address the points raised in the reviews. All changes are colored in blue.

(A) Before, we discuss the individual comments of the reviewers, we would like to clarify the positioning of the paper:

In this paper the primary contribution is the datasets and queries we executed on those datasets to construct the benchmark. Generating this benchmark is non-trivial and also required significant time, effort, and resources making it a valuable contribution to the research community as labeled training data ( which involves running thousands of queries) is very hard to find for database tasks. With this benchmark, we want to achieve two goals: (1) show that more research is needed to enable zero-shot cardinality estimation and (2) based on the dataset and workloads (with true cardinalities) enable more research in this direction.

To motivate using a zero-shot/pre-trained approach and to show that more research is needed we evaluated two model variants (one based on GNNs and transformers) in a zero-shot manner. For this we used a leave-one-out setting where we used all datasets and workloads except one for training the model and then tested on the left-out dataset and workload the zero-shot capabilities. While the models themselves are variations of standard architectures (e.g. the GNN model uses a new message passing scheme), we use a novel input representation for the datasets and queries to enable zero-shot cardinality prediction on unseen datasets and workloads.  However, the models and how we represent the datasets and workloads are only the secondary focus of this paper, and the models only serve only as the motivation to show the need for a new benchmark on zero-shot cardinality estimation and that this approach requires further research which is enabled by this benchmark.


B) Second, we want to address shared comments regarding the diversity of our datasets and workloads (vebN: W1, 4NBw:D1/D2)


The datasets and workloads (queries) of CardBech are diverse to be suitable to train and test zero-shot and pre-trained models. To quantify diversity we have updated Table 1 with additional data statistics as well as added Table 4 which summarizes the cardinalities of the workload queries. Together Table 1 and 4 illustrate the diversity of the dataset and workloads which constitute CardBench. In more detail:

1) In our original submission, Table 1 uses table count, column count, row count and unique value count as diversity metrics for the workloads.  We have updated Table 1 (in a revised submission to open review – changes are colored blue) with the number join paths (how many relationships exist between the tables of each dataset), type of join paths (star, chain, mixed) as well as the average pearson correlation between columns of the same table.
2) To quantify diversity of our workloads, we added in the appendix Table 4, that lists the max, average and standard deviation of the query cardinalities per dataset for the single table, binary join and multi-join queries. At this moment we added the statistics for the single table, binary join queries and we are calculating the statistics for the multi-join queries which we will add once we have them.

Further, we are planning to list the sources of the datasets in the repository of our benchmark (All datasets we use are publicly available). If you have suggestions for additional diversity statistics we would be happy to extend Table 1 and 4. Note that we are also providing an end-to-end training data preparation pipeline, which can be invoked on more datasets and used to generate different workloads.

---

### Comment · Area_Chair_fpfb · 2024-11-30
**The deadline for Author/Reviewer discussion period is in three days!**

Dear Reviewers,

Thanks again for providing your constructive comments and suggestions. The deadline for the Author/Reviewer discussion period is in three days (December 2). Please make sure to read the authors' responses and follow up with them if you have any additional questions or feedback.

Best,

AC

---

### Meta-Review · Area_Chair_fpfb · 2024-12-21

**Metareview:**

The paper introduces CardBench, a benchmark comprising thousands of queries across 20 distinct databases, along with scripts to compute data summary statistics and generate queries. Using this benchmark, the paper evaluates learned cardinality estimation in a zero-shot setting. GNN-based and transformer-based models are trained and analyzed under three setups: (1) instance-based, (2) zero-shot, and (3) fine-tuned.

**Strengths**
- CardBench provides a timely platform that could facilitate future research in cardinality estimation.
- The experimental results reveal some interesting insights into the challenges and performance of current approaches.

**Weaknesses**

- Reviewers are not totally convinced whether the collected datasets are comprehensive enough.
- The paper only tests typical GNN and transformer based architectures, which are insufficient. Reviewers suggest to show how other state-of-the-art methods compare on CardBench.

While CardBench has the potential to contribute to the field of cardinality estimation, the major weaknesses, including concerns about dataset comprehensiveness and insufficient evaluation with diverse methods, limit its overall impact. Without addressing these issues, the benchmark’s contribution appears to fall below the bar for acceptance.

**Additional Comments On Reviewer Discussion:**

The authors' rebuttal has addressed some of the concerns raised in the original reviews. However, several major issues remain unresolved. For instance, most reviewers expressed a desire to see how other pretrained models perform on the proposed benchmark beyond the standard GNN and transformer architectures. Including such experiments would enhance the comprehensiveness of the proposed benchmark.

---

### Decision · Program_Chairs · 2025-01-22

Reject